# Sustainability in Mountain Viticulture: Insights from a Case Study in the Portuguese Douro Region

José António Martins [1], Ana Marta-Costa [2,*], Maria Raquel Lucas [3] and Mário Santos [4,5,6,7]

[1] Department of Economy, Sociology and Management, University of Trás-os-Montes e Alto Douro (UTAD), 5000-801 Vila Real, Portugal; jamartins1960@sapo.pt

[2] Department of Economy, Sociology and Management and Centre for Transdisciplinary Development Studies (CETRAD), University of Trás-os-Montes e Alto Douro (UTAD), 5000-801 Vila Real, Portugal

[3] Department of Management, Mediterranean Institute for Agriculture, Environment and Development (MED) & Global Change and Sustainability Institute (CHANGE), University of Évora, 7004-516 Évora, Portugal; mrlucas@uevora.pt

[4] Centre for the Research and Technology of Agro-Environmental and Biological Sciences (CITAB), Institute for Innovation, Capacity Building and Sustainability of Agri-Food Production (Inov4Agro), University of Trás-os-Montes e Alto Douro (UTAD), 5001-801 Vila Real, Portugal; mgsantos@utad.pt

[5] Laboratory of Fluvial and Terrestrial Ecology, Innovation and Development Center, University of Trás-os-Montes e Alto Douro (UTAD), 5000-911 Vila Real, Portugal

[6] Department of Biology and Environment, University of Trás-os-Montes e Alto Douro (UTAD), 5000-911 Vila Real, Portugal

[7] Laboratory of Ecology and Conservation, Federal Institute of Education, Science and Technology of Maranhão, R. Dep. Gastão Vieira, 1000, Buriticupu 65393-000, MA, Brazil

* Correspondence: amarta@utad.pt

**Abstract:** Evaluations of the sustainability of the viticulture associated with wine production are still scarce in the literature. Usually, the 'carbon footprint' assumes the environmental dimension, while the economic pillar is focused on market orientation. In the present work, the integration of both was tested using a case study supported in a six-year series (2015–2021) of primary data from a farm in the Região Demarcada do Douro (Douro Demarcated Region, hereafter the RDD). Economic and environmental inputs and outputs were collected from 'pruning to harvest'. Profitability was calculated based on the market prices and environmental impacts using the AgriBalyse database, which is available in OpenLCA 1.10.3. In the scope of the economic results, the following costs stand out: (1) 'human labor'; (2) use of machinery; and (3) plant protection products. Concerning the environmental impacts, the greatest weight resulted from the use of fuels, and no relation was found between the grape production variation and environmental factor variation. The indicators studied were considered valuable for comparing production systems (conventional, organic, and biodynamic, among others) and might support stakeholders' decision making. We highlight the importance of replication in further studies to better understand the complex world of viticulture's sustainability.

**Keywords:** economic balance; environmental impacts; carbon footprint; wine grape growing production system; Douro wine region

## 1. Introduction

The impact of the wine industry on the environment is of growing interest. Studies conducted on the subject [1–5] focused on issues related to the use of natural resources, greenhouse gas (GHG) emissions, waste management, and the impact on biodiversity, supported by life cycle assessment (LCA) (e.g., [6–15]), the carbon footprint (CF) [16–23], or the ecological footprint (EF) [24,25].

LCA has allowed for the evaluation of different viticultural management approaches by comparing the results between conventional and organic systems [10,15,24,26] or biodynamic systems [11], with conventional systems generally showing higher GHG emissions

over the agricultural life cycle. However, the results obtained vary depending on the specific agricultural practices of each system [10]. Additionally, the specific geographical context and life cycle phase were considered by Laca et al. [27] as important variables for the specific analysis of the environmental impact.

The need to develop studies focused on wine LCA [2,7,8,24,28]—which includes four phases: land, viticulture, winemaking, and distribution—and especially on viticulture has been emphasized by several authors [9,10,17–20,29] in response to the scarcity of research on the subject. The viticulture phase of wine LCA is considered by Ferrara and De Feo [6] to be significantly relevant to the total environmental impacts of the sector's viticultural practices. These authors identified only five studies dedicated to 'viticulture', which hinders their comparison and the attainment of general conclusions about the environmental impacts of wine production.

More recently, the work of Letamendi et al. [9] analyzed the different stages of the life cycle of organic Chilean wine. The results remained consistent with those obtained in previous works across various parameters under analysis, with organic production proving to be environmentally and socially more favorable than conventional production.

Assessments of GHG emissions in LCA are mostly linked to CF [10,16,17,23,30]. This tool has been used to evaluate and communicate the environmental impact of products and processes in different sectors and contexts [30] and is supported in 'kg $CO_2$-Eq' [17,19,21,31], calculated using environmental equivalents defined in EcoInvent [32]. In this sense, standardizing its definition and calculation methodology is relevant to promoting transparency, consistency, and comparability of results across different studies and sectors [31], which is also considered a marketing strategy [30]. Benchmarking CF data has been reported in the wine industry in Spain [18], France [18,19], Italy [16,33], Europe [23], and the USA [20] for establishing standards and identifying best practices and emission reduction opportunities. However, only the works of Marras et al. [16] and Navarro et al. [18] focused on viticulture. In Portugal, concentrating on environmental issues and carbon and water footprints, Martins et al.'s research [28] revealed a net emission of $CO_2$ in the viticulture phase.

The CF impact per phase is associated with considerable differences [17,19,27,28,33], justified by the variability of agricultural factors [6,13,22,34], productions, designations of origin, and types of wines [17,23]. The relative importance of the 'wine LCA' varies between 9% and 50% of the total impact [16,35,36]. However, a negative value (carbon sink) has been suggested by Chiriacò et al. [5] because of sustainable viticultural practices, which is a positive sign for promoting a change in cultural practices aiming to reduce environmental impacts, considering the specific soil-climatic conditions of each farm location.

Assessment of the environmental impact of the wine industry has also been conducted through the EF, which allows identifying areas for improving efficiency and reducing the environmental impact [24], including efficient use of water resources, proper waste management, and the adoption of sustainable agricultural practices [25]. Niccolucci et al. [24] applied this specific approach to evaluate the production of two Italian wines, and more recently, Litskas et al. [25] conducted the first determination of the EF of grapes. However, several limitations have been pointed out for these works that hinder obtaining data and directly comparing results, highlighting the variety of methods and metrics [21] and lack of specific available data and their quality [24], in addition to the complexity of the wine life cycle [25], which can make it difficult to precisely quantify carbon emissions at each stage.

The requirement for multidimensional approaches has led to the combined use of LCA with other methodological instruments. Falcone et al. [37] complemented LCA with life cycle cost analysis and multicriteria analysis to include the economic aspects of wine production. Vásquez-Rowe et al. [36] investigated the environmental efficiency of grape production under different agricultural practices through LCA, and Jradi et al. [19] accomplished this through data envelopment analysis. Also, a sustainable wine scoring system (SWSS) based on the multicriteria LCA approach was proposed by Valero et al. [14].

Indeed, research on the impact of viticulture on the environment has grown in the literature (e.g., [1–5]), requiring additional dimensions of sustainability for a more com-

prehensive and balanced analysis of the sector (e.g., [38,39]). Economic and social aspects, though the latter are generally relegated to the background (Massuça et al. [40]), are the most common ones within the scope of Elkington's triple bottom line theory [41]. They contribute to a broader understanding of the challenges and opportunities associated with sustainable wine production and provide valuable insights to guide more sustainable agricultural practices in the wine sector. Studies such as those by Borsato et al. [29], Triviño-Tarradas [35], and Marta-Costa et al. [42], which focused on viticulture, and others encompassing wine production [28,34,43,44] contribute to a broader understanding of the sustainability issues in the wine industry and highlight the need for integrated and comprehensive approaches to assess and promote sustainability across the life cycle. However, many of the frameworks that have been developed converge toward a final sustainability index [42], mitigating the complexity of the sector and losing the detailed analysis of the object under review.

In this sense, the main objective of this work is to analyze viticulture from both the economic and environmental perspectives to enable a comprehensive understanding of the benefits and challenges associated with these two dimensions. From the economic dimension, this paper aims to highlight the costs and revenues generated by the activity, while from the environmental perspective, it considers different metrics whose measurements reveal the impact of the activity on the environment. Human labor is also included as an environmental factor, an approach considered scarce in the literature [11] even though it is a relevant factor, especially in mountain viticulture. Additionally, the harmonization indicators—two functional units—of kg and ha (FUkg and FUha) are used to increase the referenced intercomparability before [6,11,38] and because it reduces the variability of the environmental impact associated with the annual irregularity of grape production by using only FUkg.

Holistic approaches to the dynamics of the economic and environmental balances might support sustainable viticulture. In this context, the development of tools that conform to interventions, operations, the products used, expenses, and income will allow gaining insights on the real impacts of each viticultural campaign, which is necessary for the transition to environmentally friendly production systems. In this way, measurement of the economic and environmental balances associated with the viticultural phase of the LCA of wine production, from 'pruning to harvest', helps bridge the identified gap while serving as a starting point for future studies in this scope. This study also aims to overcome some of the identified gaps in the previous exposition through its dedication to the autonomous phase of viticulture of the LCA framework of wine production in the case of mountain viticulture (RDD) and to develop a systematic collection of quality data that allow detailed analyses and advances in data harmonization knowledge.

## 2. Materials and Methods

### 2.1. Study Area

The studied area encompasses a farm located at the western end of the Baixo-Corgo subregion of the RDD in Mesão Frio, Vila Real, Portugal (Figure 1). With an area of 3.1 hectares, it is a rain-fed farm with east and south exposure, average slopes of 30%, and 'loamy' and 'sandy loam' soil texture, subject to an average annual rainfall of 850 mm, and having an average annual sum of active temperatures 2300° C (>10 °C, from 1 January to 15 October) of 2300 °C [45,46]. The study focused on detailed data covering a series of six years to cover the diverse environmental and economic conditions of mountain viticulture in the RDD.

Farm management has been ruled in accordance with the standards of integrated crop production (ICP) [47]. In this regard, the spraying of plant protection products was carried out in accordance with the agricultural notices issued by the regional phytosanitary authorities, following the doses recommended by the manufacturers and only when necessary. Concerning fertilization, this was defined from the recommendations derived from the results of the soil and foliar analyses. A complete list of all products used (phytosanitary,

pesticides, and fertilizers) is available in the Supplementary Materials (Table S1). The vineyards were planted in the beginning of the 20th century and then replanted to meet mechanization and new technical developments around 30 years ago. Also, in agreement with ICP standards, conservation tillage was applied [48].

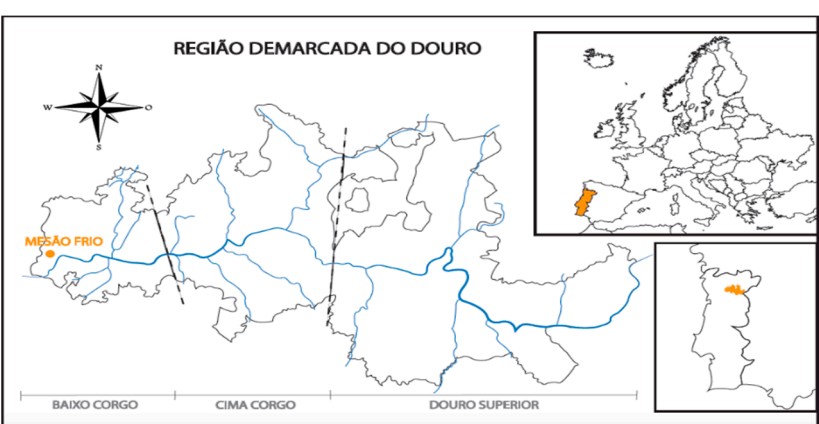

**Figure 1.** RDD map and farm localization (source: Inês Ferreira©).

### 2.2. Data Collection

The data used were limited to the viticulture phase (i.e., from 'pruning to harvest' (from November to October of the following year)), between 2015–2016 and 2020–2021. Multi-year data were considered fundamental for incorporating the gradient of economic and environmental conditions [4].

Based on previous publications [49–54], a systematic recording grid (GRS)—available in the Supplementary Materials (Table S2)—was built, including the human intervention—'interventions'—and technical work—'operations'—for each wine-growing campaign. The 'interventions' data were based on the main operations set for vineyards in RDD, whose technical details can be found in the work of Magalhães [50]: slope cleaning; digging around vines; pruning; replanting; irrigation; fertilization, herbicide application; spraying phytopharmaceuticals; grafting; bard repair; trimming; weeding; topping; redraining; technical controls; pre-harvest; harvesting; and land improvements (support walls, landings, and rainwater drainage structures).

The participation of each resource was quantified for each intervention, and the following options were considered: manual and mechanical traction as well as manual, mechanical, and electrical equipment. All materials (metal, plastic, wood, and stone, among others), products (chemical, natural, and mineral), water, and waste produced in various categories of farming activity were recorded.

### 2.3. Environmental and Economic Balances

Two functional units (FUs) were used to measure the economic and environmental balances, 'FUkg' for 1 kg of grapes produced and 'FUha' for 1 hectare of vineyard, to encompass the variability of factors [6,34], such as soil and climate, cultural practices, production methods, and grape varieties, that influence annual production [44]. Apparently, FUha is more suitable for making comparisons between farms without compromising the results obtained by FUkg [54]. FUkg served as the basis for calculating the fundamental economic and environmental balances for each campaign. FUha was used to compare the real impacts on the vineyard as a whole, both economic and environmental, over the series being analyzed.

#### 2.3.1. Environmental Assessment

Environmental impact calculations were carried out using the AGRIBALYSE Life Cycle Inventory (LCI) Database (1.10.3) available on OpenLCA [25]. For the specific N calculations

($NO_3$, $N_{org}$, $N_2O$, $NO_x$, and $NH_3$), the formulae defined by Nemecek and Schnetzer [55] and Nemecek et al. [49] were used. A list of the inputs and outputs considered in the environmental assessment is available in the Supplementary Materials (Table S3).

Although most studies tend to emphasize the CF for an environmental approach [9,10,23,30], some authors considered it relevant to broaden the set of results with the actual environmental impact [11]. The following indicators were considered: marine eutrophication; freshwater eutrophication; acidification; human toxicity (via air); human toxicity (via soil); ionizing radiation; ozone layer depletion; and global warming potential (GWP) at 20, 100, and 500 years.

These indicators, apart from being regularly cited [9,11,26,27,31], are easy to link with viticulture (N, P, and SO), and they are related to environmental issues (global warming, the ozone layer, and toxicity). Additionally, the combination of indicators enables a more assertive evaluation of the environmental impacts of viticulture and comparisons [11]. Assessments of these indicators, whenever possible, were made using more than one impact estimation method: EDIP; EDIP 2003; RECIPE MidPoint (H) v1.13; CML 2001; ILCD 2.0-2018 Midpoint; and IPCC 2013, in line with Silva et al. [56]. The GWP at 100 years was considered the broadest in terms of reporting.

Land mobilization was not considered as the crop is permanent and has been established for over 100 years. The last soil mobilization to restructure the vineyard took place over 30 years ago, well above the 20 years considered relevant by Nemecek et al. [49], which is based on the IPCC 2006 methodology [57].

Environmental risk of phytopharmaceuticals, herbicides, insecticides, and fertilizers applied was also disclosed (from the AGRIBALYSE Database). The assessment methods and indicators extracted are shown in Table 1. The criteria for the choice were (1) the use of assessment methodologies available on Open LCA; (2) the used impact indicators that were present in at least two methods with the same functional units of measurement; and (3) being representative of the main perceived environmental impacts.

**Table 1.** Average environmental impact values by six different environmental assessment methods for the wine-growing seasons between 2015–2016 and 2020–2021.

| | | Environmental Impact | | | | | | | | |
| | | Methods | | | | | | Max | Min | Standard Deviation |
| Indicators | Unit | EDIP | EDIP 2003 | ReCiPe MidPoint (H) v1.13 | CML 2001 | ILCD 2.0—2018 Midpoint | IPCC 2013 | | | |
|---|---|---|---|---|---|---|---|---|---|---|
| Eutrophication Marine | kg N-Eq | - | - | $1.742 \times 10^{-1}$ | - | $1.703 \times 10^{0}$ | - | $1.742 \times 10^{-1}$ | $1.703 \times 10^{0}$ | $7.652 \times 10^{1}$ |
| Eutrophication Freshwater | kg P-Eq | - | - | $1.371 \times 10^{-2}$ | - | $1.368 \times 10^{-2}$ | - | $1.371 \times 10^{-2}$ | $1.368 \times 10^{-2}$ | $1.535 \times 10^{-5}$ |
| Acidification | kg $SO_2$-Eq | $3.071 \times 10^{-1}$ | - | $2.524 \times 10^{-1}$ | $3.071 \times 10^{-1}$ | - | - | $3.071 \times 10^{-1}$ | $2.524 \times 10^{-1}$ | $2.577 \times 10^{-2}$ |
| Ionizing Radiation | kg U235-E | - | - | $4.358 \times 10^{-1}$ | $4.354 \times 10^{-1}$ | - | - | $4.358 \times 10^{-1}$ | $4.354 \times 10^{-1}$ | $2.023 \times 10^{-4}$ |
| Ozone Layer Depletion | kg CFC-11-Eq | $8.724 \times 10^{-3}$ | $9.207 \times 10^{-3}$ | $1.003 \times 10^{-2}$ | $9.207 \times 10^{-3}$ | $9.896 \times 10^{-3}$ | - | $1.003 \times 10^{-2}$ | $8.724 \times 10^{-3}$ | $4.835 \times 10^{-4}$ |
| Human Toxicity (via Air) | $m^3$ Air | $1.379 \times 10^{5}$ | $1.375 \times 10^{5}$ | - | - | - | - | $1.379 \times 10^{5}$ | $1.375 \times 10^{5}$ | $2.169 \times 10^{2}$ |
| Human Toxicity (via Soil) | $m^3$ Soil | $8.398 \times 10^{-2}$ | $8.398 \times 10^{-2}$ | - | - | - | - | $8.398 \times 10^{-2}$ | $8.398 \times 10^{-2}$ | $0.000 \times 10^{0}$ |
| GWP 20y | $CO_2$-Eq | $4.988 \times 10^{-1}$ | $4.237 \times 10^{-1}$ | - | $5.063 \times 10^{-1}$ | - | $5.008 \times 10^{-1}$ | $5.063 \times 10^{-1}$ | $4.237 \times 10^{-1}$ | $3.399 \times 10^{-2}$ |
| GWP 100y | $CO_2$-Eq | $5.024 \times 10^{-1}$ | $4.265 \times 10^{-1}$ | $5.015 \times 10^{-1}$ | $5.025 \times 10^{-1}$ | $5.046 \times 10^{-1}$ | $4.884 \times 10^{-1}$ | $5.046 \times 10^{-1}$ | $4.265 \times 10^{-1}$ | $2.786 \times 10^{-2}$ |
| GWP 500y | $CO_2$-Eq | $4.299 \times 10^{-1}$ | $3.662 \times 10^{-1}$ | - | $4.273 \times 10^{-1}$ | - | - | $4.299 \times 10^{1}$ | $3.662 \times 10^{-1}$ | $2.945 \times 10^{-2}$ |

### Human Labor and Its Potential Environmental Impact

Given the human labor costs reported for RDD [58] and confirmed with our assessment, we decided to estimate its environmental impact independently using the methodology proposed by Rugani, Pananiuk, and Benedetto [59], the later of whom used it in a comparative LCA for the ecological footprint between viticultural methods [11]. Human labor is divided into three categories: qualified worker (HL-1), technician (HL-2), and manual worker (HL-3). Through the ReCiPe MidPoint (H) method, results were extracted for 14 of the 15 environmental impact indicators included. In our assessment, we used the category technician (HL-2) and only the environmental indicator of the GWP for 100 years.

2.3.2. Economic Assessment

The economic calculations were gauged in Euros (EUR) at 'current prices', which were standardized to constant prices for 2021 according to the data provided by the Portuguese Statistics Institute (INE) [60].

'Expenses' correspond to the real market costs invoiced and paid in each campaign and those defined administratively by the Portuguese state, such as the minimum wage for agricultural workers and the employer's compulsory social benefits paid.

The allocation of incurred costs took into account the different factors of production used, such as (1) human labor (hourly rate based on the Portuguese minimum wage, including the cost of social benefits paid by the employer); (2) mechanical traction (rent/hour only for the machine without an operator and without fuel); (3) phytopharmaceuticals (herbicides, fungicides, and insecticides) and fertilizers; (4) fuel (diesel and petrol); (5) insurance (harvest, labor, and machinery) and taxes (real estate); (6) administrative and technical costs (technical assistance, ICP certification, association fees, and soil, foliar, and ripeness analyses); (7) depreciation (mechanical and manual tools and plants, allocated on the basis of the depreciation percentages legally defined in Portugal); (8) other (piped water); and (9) opportunity costs (with a 1% underlying rate).

All amounts paid for goods and services purchased included VAT. Revenue was calculated based on the price paid for the grapes by the local winery per season (and the market value of the woody matter removed from the vineyard for burning). The value of the 'subsidies' received was awarded by the Portuguese administrative body, which grants support to agriculture (Instituto de Financiamento Agricultura e Pescas (IFAP), Agriculture and Fisheries Financing Institute) [61]. Income taxes associated with operations were not included, as they depend on the set of economic activities carried out by the winegrower.

## 3. Results

The results are presented in two sections—economic and environmental—and within the former, there is a specific division of human labor due to its particularities [11,59].

### 3.1. Economic Assessment

The results obtained were significant in identifying the main factors that contributed to the balance of each campaign and the series under study (Figure 2).

Human labor was by far the most important factor in terms of costs, always accounting for more than half of the total expenditure on each campaign, with an average for the period of 53.60% (standard deviation of 3.12%). This was followed by mechanical traction (without fuel) with an average of 17.23% expenditure (standard deviation of 2.08%).

The remaining expenses were as follows (mean, standard deviation): insurance and taxes (10.07%, 1.67%); phytopharmaceuticals and fertilizers (9.73%, 2.67%); administrative and technical (4.74%, 0.76%); fuel (2.55%, 0.21%); depreciation (1.08%, 0.29%); opportunity costs (0.99%, 0.0%); and other (0.01%, 0.0%). It should be noted that mechanical traction, together with fuel, accounted for close to 20% of the annual expenditure and human labor accounting for 75%. On the other hand, the sum of the factors (mechanical traction, fuel, phytopharmaceuticals, and fertilizers) represented 29.51% of the total costs. However, when the human labor factor (in the vineyard) was considered, the relative economic impact rose to 83.11%.

Different seasons (Figure 3) were associated with variations in the total expenditures, income and subsidies, and grape production (kg) per hectare. The annual costs also varied annually due to the intensity of labor factors (human and machine) and products applied in phytosanitary treatments, which are justified and conditioned by the soil and climate conditions [6,22,34]. The economic balance was negative (EUR −468.5 per hectare) on average for the period but became positive (EUR 493.8 per hectare) when considering the subsidies received by the farm through agri-environmental measures to support maintenance of the walls and landings.

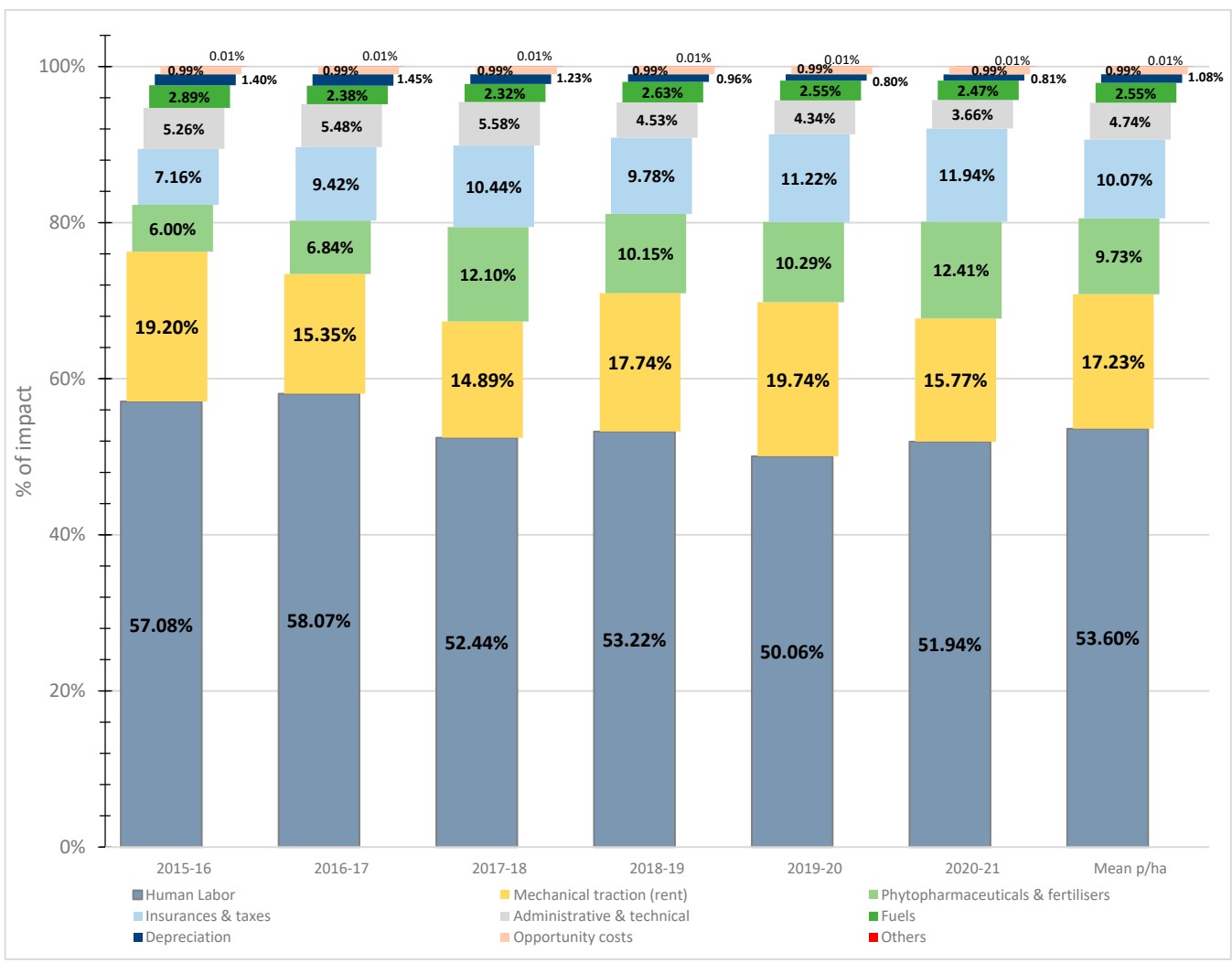

**Figure 2.** Proportions of different cost categories (human labor, mechanical traction, phytopharmaceuticals and fertilizers, insurances and taxes, administrative and technical issues, fuels, depreciation, opportunity costs, and other) in total production costs and the average for the wine-growing seasons between 2015–2016 and 2020–2021.

Yields were structurally low when compared to other wine-growing regions [62] but above the average for the RDD and the Baixo Corgo subregion [49]. Farmers are always seeking cost reductions in all interventions and operations, essentially due to the price paid for grapes. Harvesting, pruning, work to maintain the land's support structure (walls, slopes, and landings) and the vines' vegetation wall (trimming and pruning) and maintaining the vegetation cover between the rows (weeding) accounted for around 70% of the total human labor employed (Figure 4). There were no studies that could be compared in this matter, but the experience perceived pointed to a similar situation in all of the RDD.

The overall costs allocated to human labor per hectare for each season and the estimated average values involved in the series showed a sharp reduction in the first half of the series due to the substantial decrease in the number of hours allocated (Figure 5).

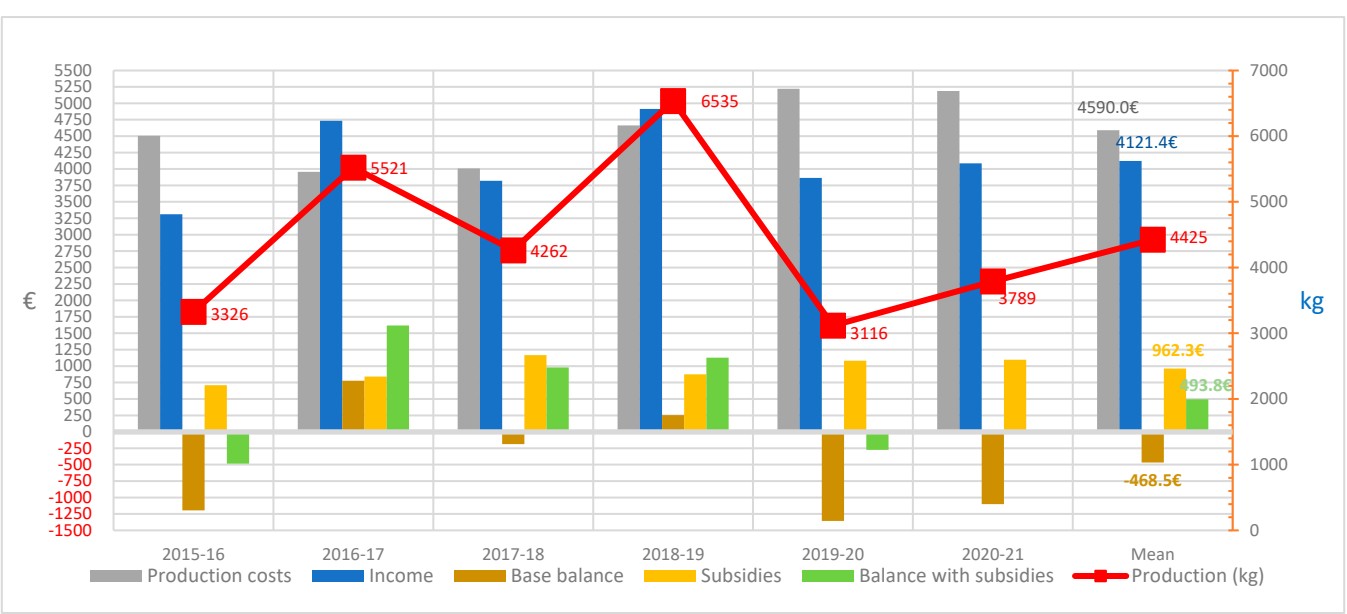

**Figure 3.** Evolution of production costs and incomes and their balance with and without subsidies, in EUR and grape production (kg) per hectare for the wine-growing seasons between 2015–2016 and 2020–2021.

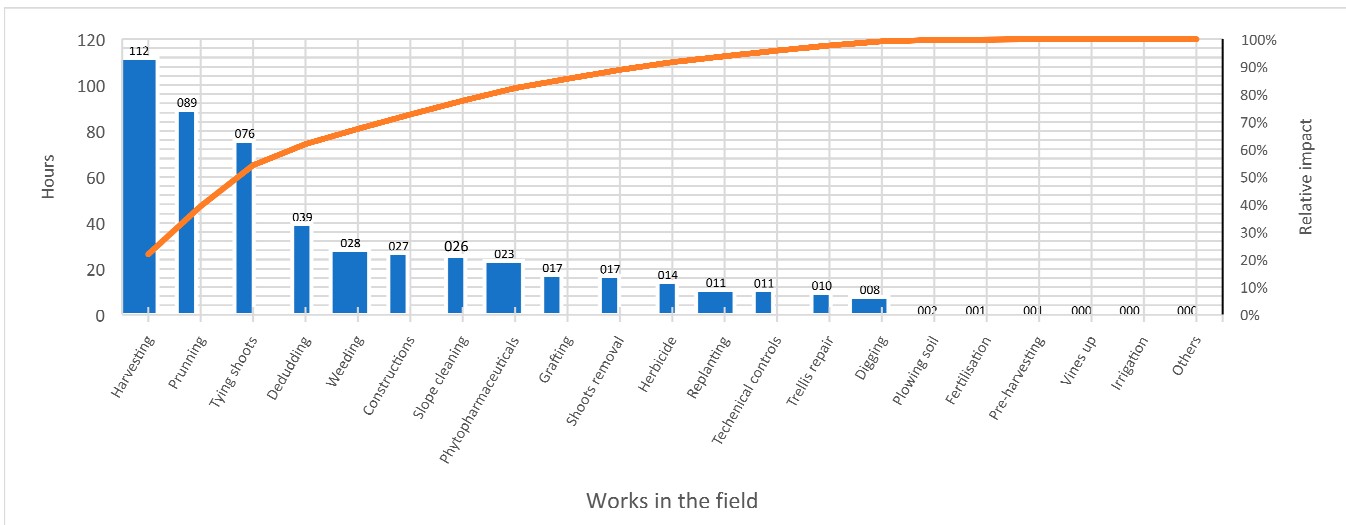

**Figure 4.** Average human labor used per operation in hours per hectare for the wine-growing seasons between 2015–2016 and 2020–2021, and the correspondent Pareto's Line.

In the second half of the series, the costs showed a growth trend despite the stabilization of the number of working hours, which was associated with the base value of the national minimum wage. The average period of hours worked per hectare was slightly higher (from 5 to 7%) than the information perceived by the RDD, but without similar studies, we could not compare the results.

*3.2. Environmental Assessment*

3.2.1. Evaluation Methods and Indicators

Eight indicators were considered, while the GWP was divided into three different time periods of impact (20, 100, and 500 years) (Table 1). However, only the results for the GWP over 100 years are presented in the following figures.

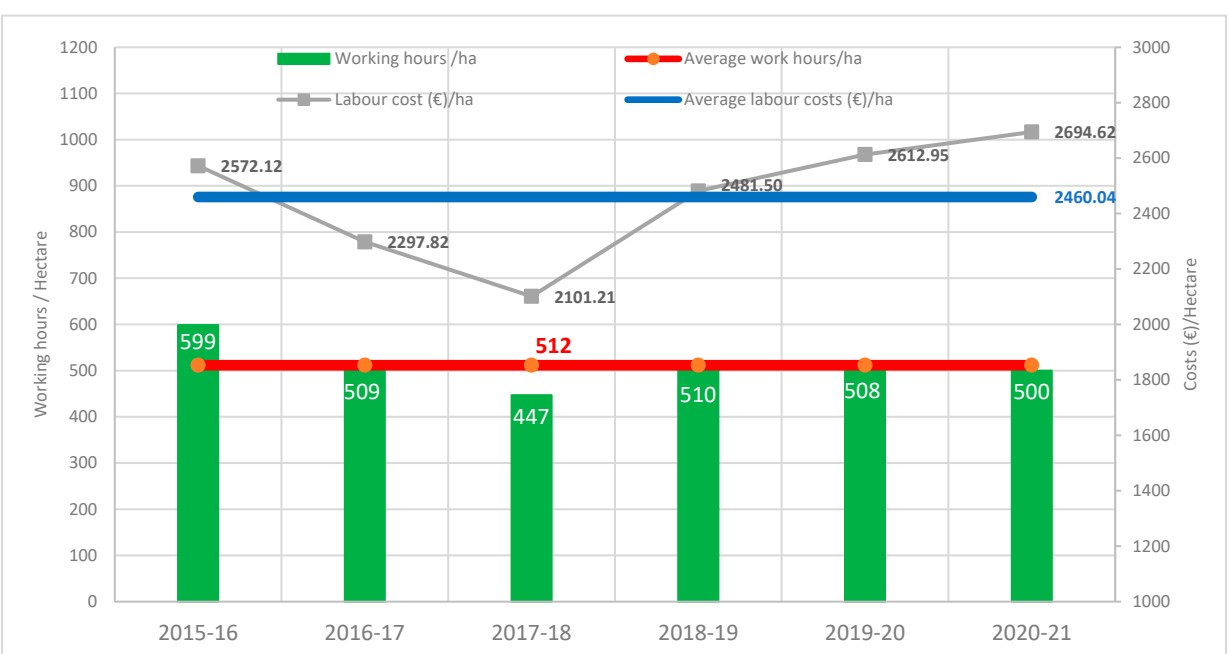

**Figure 5.** Evolution of human working hours and labor costs in EUR for the wine-growing seasons between 2015–2016 and 2020–2021.

Figure 6 shows the active ingredients used in the series (2015–16 to 2020–21) and the distribution of their relative importance. In absolute terms, sulfur was by far the most widely used ingredient, followed by folpet, copper, mancozeb, and metalaxyl, all of which are active ingredients intended for the prevention and treatment of mildew (*Plasmora viticola*) and powdery mildew (*Erysiphe necator*). The number of applications required varies according to the meteorological conditions [2,12,21]. Glyphosate, used to combat weeds, is applied once per season, namely on slopes and between lines. The Supplementary Material (S1) shows all the product and ingredient quantities used each year per hectare.

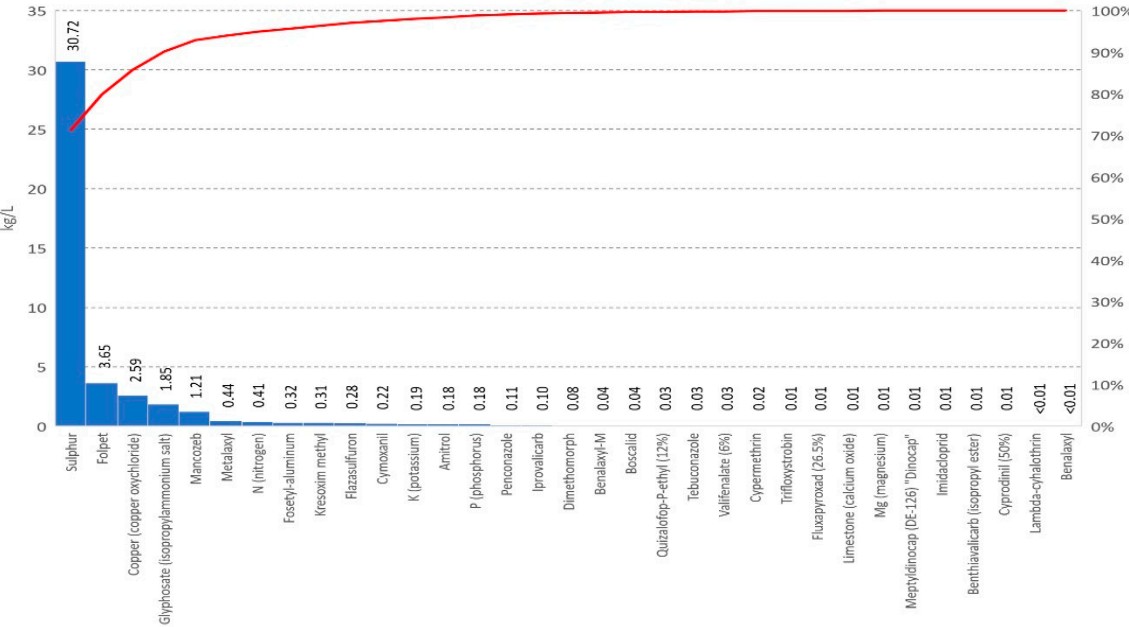

**Figure 6.** Average phytopharmaceutical active ingredients used in kg/L per hectare for the wine-growing seasons between 2015–2016 and 2020–2021 and the correspondent Paretos's Line.

Figures 7 and 8 highlight the environmental indicators associated with grape production per hectare. Figure 7 presents a set of seven environmental indicators that met the criteria of choice outlined and shows the average results obtained by each environmental method used. The average results did not show a direct relationship with the annual yield. Figure 8 highlights the GWP over 100 years (kg $CO_2$-Eq) in relation to the annual yield. There was consistency in the values presented by each environmental method used and for each year analyzed, but once again, a direct relationship with grape production was not found.

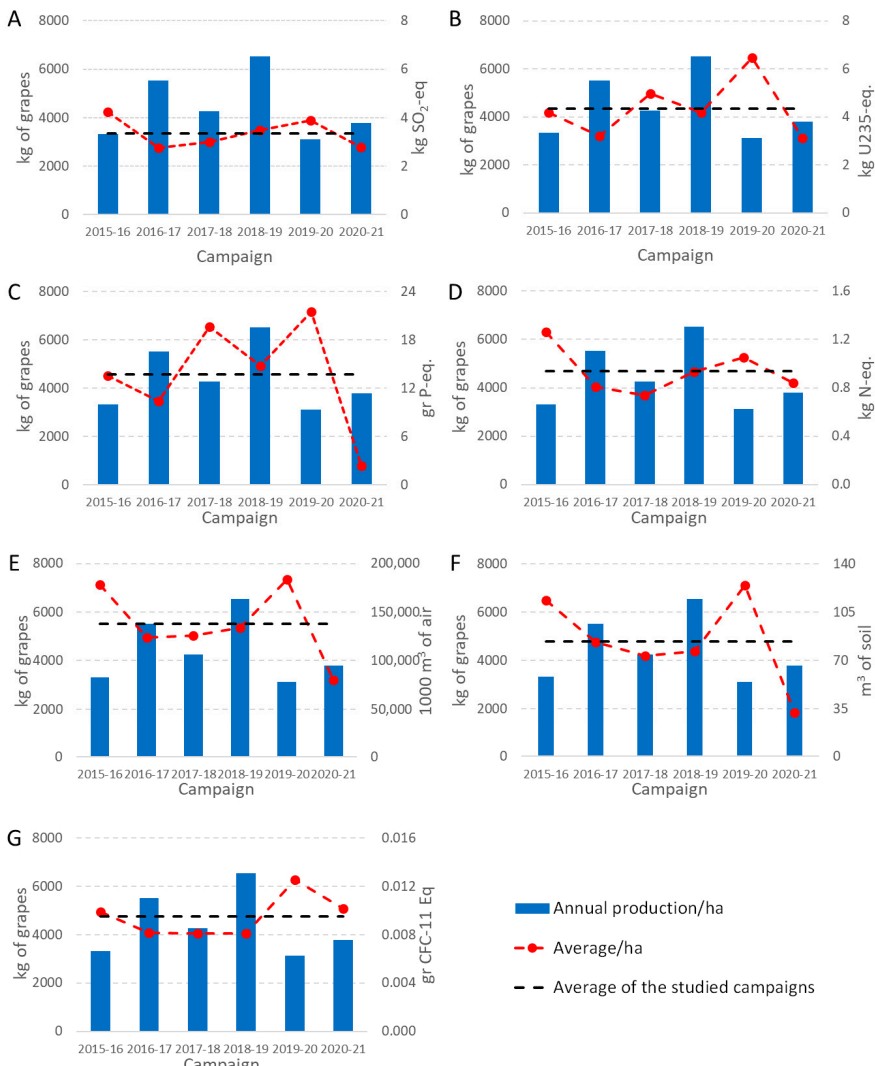

**Figure 7.** Environmental indicators vs. annual grape production (kg/ha) for the wine-growing seasons between 2015–2016 and 2020–2021: (**A**) acidification (kg $SO_2$_Eq); (**B**) ionizing radiation (kg U235_Eq.); (**C**) fresh water eutrophication (kg P_Eq.); (**D**) marine eutrophication (kg N_Eq.); (**E**) human toxicity ($m^3$ Air); (**F**) human toxicity ($m^3$ Soil); and (**G**) ozone layer depletion (kg CFC-11_Eq.).

The relationship between the grape production per hectare, annual quantity of products used, and number of spraying interventions per year is depicted in Figure 9. The number of sprays is primarily related to the meteorological conditions recorded, and each year and showed an increasing trend. In the second part of the series, there was an increase in the number of pesticides used, which was due to compliance with rules that do not allow the use of certain chemical groups more than two or three times a year. The results did not show a direct relationship between the two.

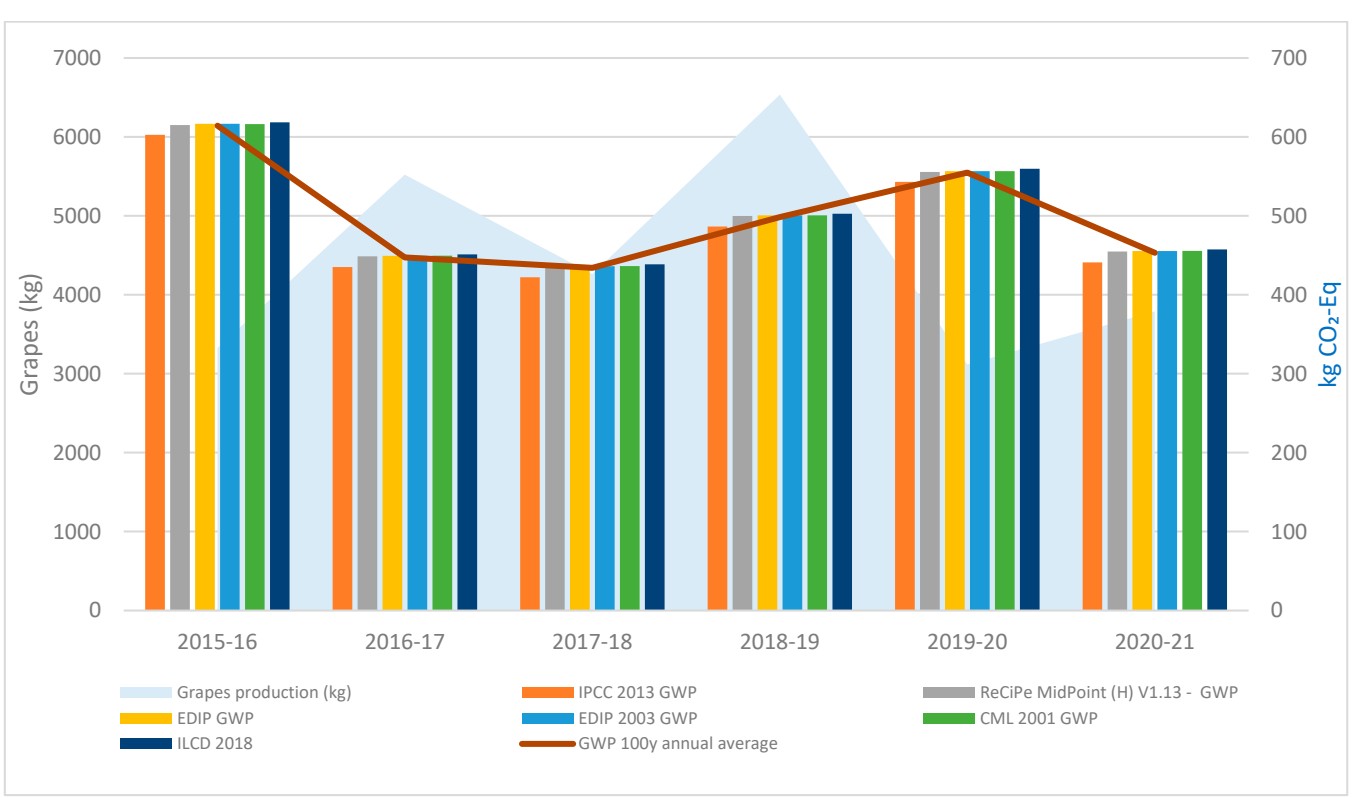

**Figure 8.** Evolution of grape production vs. GWP over 100 years (kg $CO_2$-Eq) per hectare using the six selected environmental methods for the wine-growing seasons between 2015–2016 and 2020–2021.

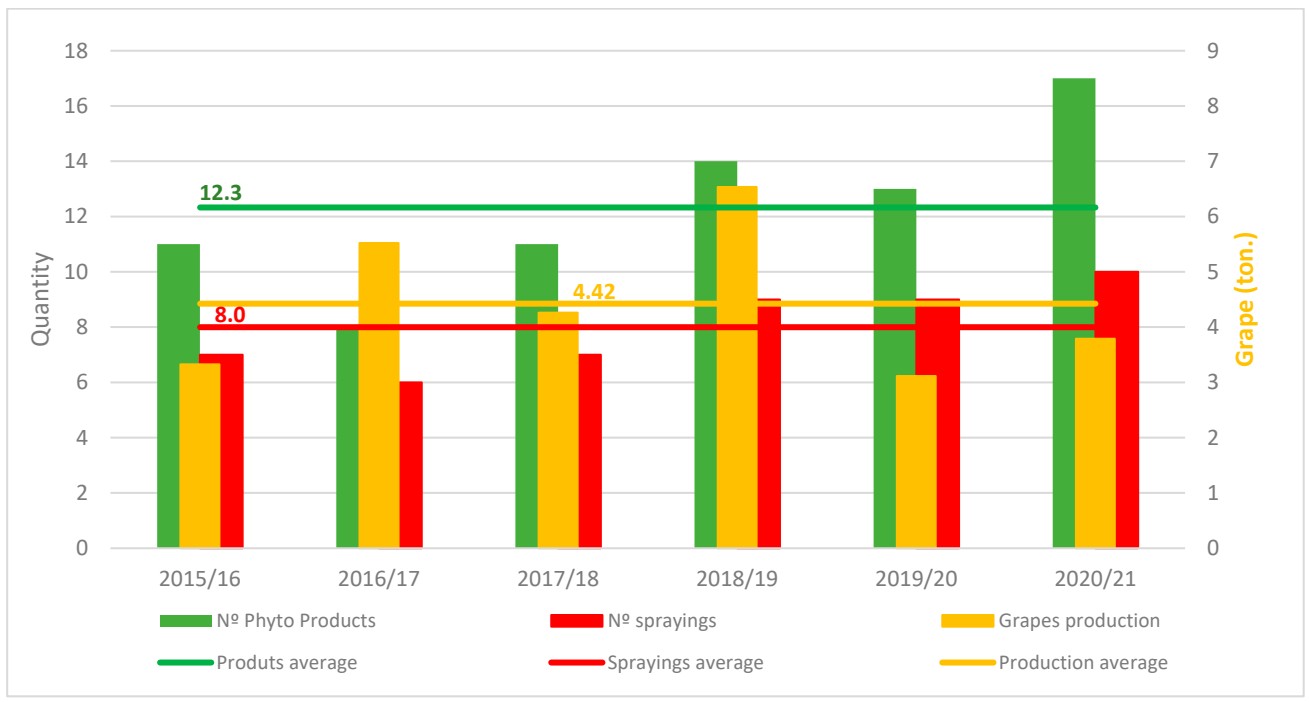

**Figure 9.** Evolution of the annual quantity of phytopharmaceuticals products used, the number of spraying interventions per year, and grape production per ha for the wine-growing seasons between 2015–2016 and 2020–2021.

### 3.2.2. Human Labor and CO$_2$-Eq

We considered HL-2 [59] to be representative of the workers and only used the CO$_2$-Eq. indicator. An impact of 0.46 kg CO$_2$-Eq per hour and 235.51 kg CO$_2$-Eq per hectare was found (Figure 10). The results showed a direct relationship with human labor costs. However, as shown in Figure 11, in the overall assessment, there was a significant difference in impact on the costs and GWP. Human labor represented more than 50% of the costs and only one-third of the GWP impact.

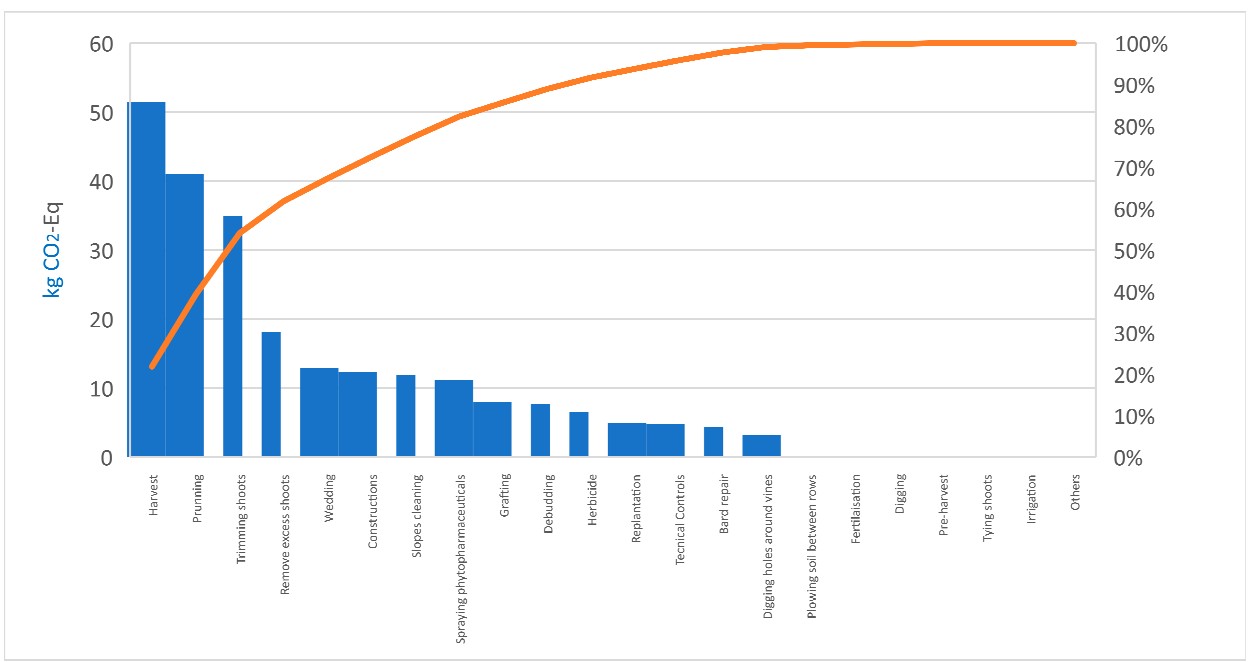

**Figure 10.** Average distribution of human labor per intervention (%) and CO$_2$-Eq impact in kg per hectare for the wine-growing seasons between 2015–2016 and 2020–2021 and the correspondent Pareto's Line.

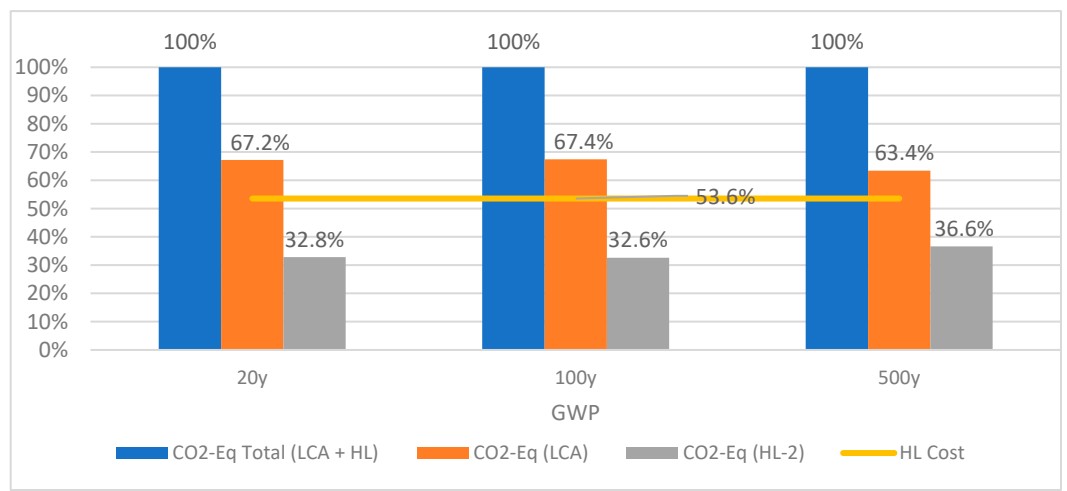

**Figure 11.** Comparison between percentage of impact kg CO$_2$-Eq and human labor cost (hectare).

The annual HL-2 impacts and the GWP at 20, 100, and 500 years associated with grape production are shown in Table 2.

**Table 2.** GWP at 20, 100, and 500 years (in kg CO$_2$-Eq) annual average values (between 2015–16 and 2020–2021) per hectare and kg of grapes, with LCA and HL-2 assessed.

|  | **GWP 20y** | | **GWP 100y** | | **GWP 500y** | |
|---|---|---|---|---|---|---|
|  | **Hectare** | **Kg** | **Hectare** | **Kg** | **Hectare** | **Kg** |
| LCA average | 482.38 | 0.109 | 487.63 | 0.110 | 407.80 | 0.092 |
| HL-2 | 235.51 | 0.053 | 235.51 | 0.053 | 235.51 | 0.053 |
| Total | 717.89 | 0.162 | 723.14 | 0.163 | 643.31 | 0.145 |

The CO$_2$-Eq results, including HL-2, show that the contribution of human labor varied between 32.6% (GWP 100 y) and 36.6% (GWP 500 y) (i.e., an average economic impact of 54% of the costs incurred per season (Figure 11)). Human labor is not only economic relevant but also has an important environmental contribution.

*3.3. General Overview Assessment*

The global overview (Figure 12) of the results obtained in this study per hectare, both economic (in EUR) and environmental (using IPCC 2013 GWP 100y in kg CO$_2$-Eq), showed a negative operating balance that only became positive with the inclusion of subsidies but still with low profitability. GWP 100y depicted a difference between the total CO$_2$-Eq reported without human labor and with HL-2 included, representing an increase of 48.22% in emissions. Despite the annual variations found in grape production and the economic balance, the GWP showed a highly stable trend.

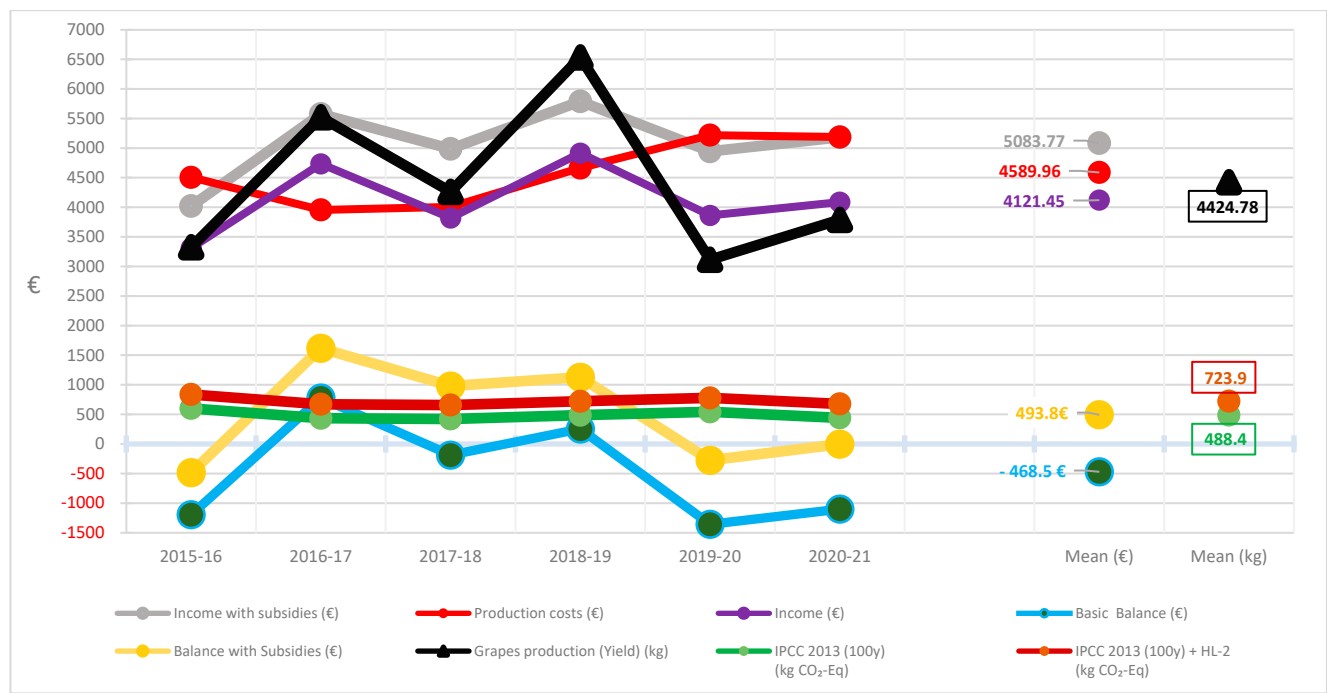

**Figure 12.** Evolution per hectare of production (kg), expenses, and revenues and its balances in EUR and CO$_2$-Eq emissions (kg) for the wine-growing seasons between 2015–2016 and 2020–2021.

## 4. Discussion

Agriculture has been shifting its paradigm from productivity to sustainability, and this should be evaluated while considering not only the economic but also the social and environmental dimensions [63]. The wine sector is following this trend [3], using the sustainability footprint concept as well as LCA. Anyway, due to the multiplicity of factors 'from the field to the bottle', they still need to be fine-tuned so that they can be validated and accepted as reliable measures [25,35,37,43].

Diverse criteria and data associated with geography, years of production, meteorological conditions, cultural practices, soils, and grape varieties, among other factors, particularly in the viticulture phase, do not allow straightforward comparisons [4,17,64,65]. This also reduces the possibility of linking production methods with environmental gains (or losses) [66]. Due to this, the development of similar approaches for data collection was one of the ideas of our work. The difficulty of obtaining primary data, either because they do not exist or because they are incomplete, constitutes one of the drawbacks of winegrowing sustainability-related studies [6,13,15]. To overcome this problem and test methodologies, a case study which included a series of years allowed detailed analysis of the economic and environmental components of wine campaigns [38].

The segmentation of economic and environmental factors was considered fundamental to tackle the relevance of each when evaluating the sustainability of a product (the social dimension in the case of human labor). Furthermore, an in-depth analysis of the factors involved in grape production linked with carbon balances is particularly important for farmers accessing the financial markets that trade $CO_2$-Eq emission rights. Our results point to the need for an accurate evaluation of the sustainability of the wine sector, splitting up the different phases to inform the consumer of the real impact of each one. Also, in our case study, ideas concerning the importance of the viticulture phase in the wine LCA were discussed.

### 4.1. Economic Assessment

The findings confirm the low yields per hectare, although they were above the average for the RDD [62]. This is a common balance in the region [38]. This outcome is related to the low price paid for grapes when considering the production costs [58]. A farm achieves profitability when the subsidies paid to the farm are added, namely those relating to compliance with agri-environmental measures for maintenance of the stone walls [61]. These features are relevant to the World Heritage landscape classification by UNESCO [67]. There is an ongoing debate in the RDD about the 'fair price' to be paid for grapes which are controlled designation protected (Denominação de Origem Controlada (DOC)) wines, DOC Douro, and DOC Porto, and thus this contribution can help to find a path that leads to economic and social sustainability of the region's wine farms and others in Portugal.

### 4.2. Environmental Assessment

The systematic collection of primary data related to the operations and interventions carried out in a vineyard enabled a determination of the factors of the environmental impact within each viticultural campaign [6]. The variability of environmental factors conditioning the winegrowing in the literature should be understood not as a problem but as the characteristics of each wine-growing farm [22,34].

The main environmental impacts, transversal to the results reported by the different assessment methods used, were fuel and plant protection products. They were in line with what has been reported in the literature for the viticulture phase [6,22,25], except for 'ground mobilization' [48]. The environmental impact methodologies using indicators were able to help in the comparisons, revealing a rather significant coherence of values. The exception was the marine eutrophication indicator, which showed a considerable difference ($10\times$) between its minimum value (ReCiPe MidPoint (H) v1.13) and its maximum value (ILCD 2.0—2018 Midpoint). Anyway, a study with the environmental footprint method published by Litskas et al. [25] and also through OpenLCA reported a value slightly above the average found here.

The problems associated different methods and software were already referenced [56], finding inconsistencies and discrepancies, some significant, between different indicators for the same system as well as for the use of different LCA assessment software. Additionally, most of the measurements in the different studies were carried out for the LCA of wine as a whole. Because of this, comparisons of the results obtained with other research were not evident, as is commonly reported in the literature [4,17,64].

The exception is $CO_2$-Eq, a unit of measurement associated with the GWP which, despite the data limitations mentioned above, is the factor with the greatest visibility and public acceptance broadly [10,16,26,27,30].

The findings point to values per FUkg/grape being relatively below those of other similar evaluations [6,22,35,68]. However, as those authors referred to earlier, the limits, objectives, and perspectives of evaluation were quite diverse, and most of them were not exclusively focused on viticulture. A similar finding is associated with FUha; it does not allow satisfactory comparison, since it is not commonly reported.

### 4.3. Human Labor

Thus far, only one study has been found within the scope of the LCA of wine and vineyards focusing on the environmental impact of human labor [11]. Villanueva-Rey et al. [11] used the assessment methodology proposed by Rugani, Panasiuk, and Benetto [59]., which quantifies the environmental impact of human work using the ReCiPe MidPoint (H) method. The actual results point to an environmental impact (GWP 20y, 100y, and 500y) that varied between 33% and 37% (in addition to the total results obtained with all other factors). This increase was roughly close to the percentage value reported by all environmental impact methods used in the study. On the other hand, the value assessed in this work was higher than the value reported by Villanueva-Rey et al. [11], probably due to a smaller number of hours per hectare or the HL category used to assess the shown results [59].

Nevertheless, the need for including the environmental assessment of human labor in the calculations of any LCA is fundamental, namely within interventions that are not easily replaced by machines (pruning, trimming shoots, grafting, debugging, and others).

### 4.4. Functional Units

The yield and environmental impacts seemed uncorrelated, showing the importance of using the two FUs (hectare and kg) to measure economic and environmental factors and address balances. The FUha results confirm the literature's recognition [6,11,38] of its suitability to informing about the great variability of factors that influence annual production in viticulture [44]. The great variability in the annual yield complicates comparability, particularly in terms of the perception of the environmental impact of farms [38,64].

## 5. Conclusions

This work took an exclusive approach to viticulture, which makes it unique in the context of works involving wine LCA in the RDD. Its objectives were to discuss issues associated with the economic and environmental impacts, informing the other phases of wine LCA but also new possibilities in terms of assessing their respective impacts.

Its innovation also comes from identifying the economic and environmental impacts of human labour as factors to be considered in production process. It lays the foundation for extracting the inherent social impacts, including the human labor required, the income that can be obtained from it, and the potential impacts on workers' health. It also opens space for new lines of research to further develop and improve environmental impact assessment methods. Additionally, the findings obtained can support techniques to reduce environmental impacts, namely by exploring the potential causal relationships between the environment and economy, in order to find a 'fair price' for grapes while contributing to a more sustainable viticulture.

This study's limitations are related to the fact that only one case study was used due to the detail and volume of information required. To overcome this issue, we consider fundamental testing of the method in other farms (and regions) and farmers' literacy concerning environmental and economic balances, which will be essential tools to support more efficient decision making.

**Supplementary Materials:** The following supporting information can be downloaded at https://www.mdpi.com/article/10.3390/su16052050/s1. Table S1: Active Ingredients; Table S2: Registration Grid; Table S3: Inputs-Outputs.

**Author Contributions:** Conceptualization, J.A.M.; methodology, J.A.M.; validation, J.A.M., A.M.-C., M.R.L. and M.S.; formal analysis, J.A.M.; investigation, J.A.M.; resources, J.A.M.; data curation, J.A.M.; writing—original draft preparation, J.A.M.; writing—review and editing, J.A.M., A.M.-C., M.R.L. and M.S.; supervision, A.M.-C., M.R.L. and M.S.; project administration, J.A.M., A.M.-C., M.R.L. and M.S.; funding acquisition, J.A.M., A.M.-C., M.R.L. and M.S. All authors have read and agreed to the published version of the manuscript.

**Funding:** This research was supported by national funds through the FCT (Portuguese Foundation for Science and Technology) under projects UIDB/04011/2020 (https://doi.org/10.54499/UIDB/04011/2020), UIDB/05183/2020 (https://doi.org/10.54499/UIDB/05183/2020; https://doi.org/10.54499/LA/P/0121/2020), and UIDB/04033/2020 (https://doi.org/10.54499/UIDB/04033/2020).

**Informed Consent Statement:** Informed consent was obtained from all subjects involved in the study.

**Data Availability Statement:** Data are unavailable due to privacy or ethical restrictions.

**Acknowledgments:** We would like to acknowledge the support by the R&D project 'Vine & Wine PT—Driving sustainable growth through smart innovation', with the financial support of the PRR (Recovery and Resilience Plan) and EU Next Generation funds, under the auspices of the 'Agenda for mobilising reindustrialisation'. We would also like to acknowledge the Projects Innovative concepts and technologies for ECOlogically sustainable NUTRIent management in agriculture, aiming to prevent, mitigate, and eliminate pollution in soils, water, and air (ECONUTRI) under the grant Horizon/101081858 and Integrated SERvices supporting a sustainable AGROecological transition (AGROSERV) under the grant Horizon/101058020. The authors would like to thank Pedro Kendall for his cooperation in consolidating the systematization of cultural operations and interventions, Thyago do Carmo Brito for reviewing the environmental inputs and outputs and validating the results extracted from the different environmental validation methods used, and Arnaldo Queirós and Humberto Coutinho for their support and field demonstrations of the cultural practices carried out on the farm.

**Conflicts of Interest:** The authors declare no conflicts of interest.

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
