# Peer review of "Sustainability in Mountain Viticulture: Insights from a Case Study in the Portuguese Douro Region"

_sustainability, doi:10.3390/su16052050_

Round 1

Reviewer 1 Report (Previous Reviewer 1)

Comments and Suggestions for Authors

Nothing to add (to previous report)

Comments on the Quality of English Language

Nothing to add (to previous report)

Author Response

Thank you very much for your appreciation.

Reviewer 2 Report (Previous Reviewer 2)

Comments and Suggestions for Authors

Dear Authors,

Interesting results are well presented. The description of the work is acceptable. The length of the manuscript is appropriate. Discussion and conclusion is detailed. In my opinion this manuscript can be PUBLISH in Sustainability especially considering the scope and topics of this journal. The authors correct all suggestions that reviewers gave about article.

I wish a lot of success to the authors.

Regards!

Reviewer

Author Response

Thank you very much for your appreciation.

Reviewer 3 Report (Previous Reviewer 3)

Comments and Suggestions for Authors

The current version of the manuscript has seen significant improvement compared to the previous one. I believe the authors have conducted thorough research, and I appreciate their efforts. However, there is still room for improvement in the writing style. I encourage the authors to share the content with friends outside the academic circle and adjust sentence structure, grammar, and the order of arguments based on their feedback, in order to tell the story more effectively and clearly.

Author Response

Thank you very much for your kind and thoughtful comments, which we believe have helped us in improving the article. We tried to increase the writing style in the version submitted at 19 of February (by email) and in this last version.

Reviewer 4 Report (Previous Reviewer 4)

Comments and Suggestions for Authors Authors did revision well.   Congratulations

Author Response

Thank you very much for your appreciation.

Reviewer 5 Report (New Reviewer)

Comments and Suggestions for Authors

       The study area was 3.1 hectares, and the large size indicators such as Human Toxicity (via Air), Human Toxicity (via Soil), Ionising Radiation,Ozone Layer Depletion and Global Warming Potential were assessed, whether the analysis of the relationship between the both was reasonable? I think some other micro-environmental indicators needs be focused.

       The data was delt with in the method of description, so the conclusions might be unreliable.         In general, I think the design of experiment is unreasonable, so I suggest to reject the manuscript.

Author Response

Thank you for your comments and for the time dedicated to our work. We agree with the observation made regarding the choice of indicators. However, the selection of the 8 indicators (out of 184 possible indicators) was due to the fact that, as we mentioned in the text, they are the ones that, out of the 6 analysis methods used, manage to meet the minimum comparability criterion in order to be used in future assessments. In this sense, these indicators allow for the establishment of easily comparable reference standards: a) due to the broad environmental scope they address; b) the unit in which they are expressed; c) and they are present in at least two of the 6 methods used.

Round 2

Reviewer 5 Report (New Reviewer)

Comments and Suggestions for Authors

      I'm glad my suggestion can help you.

This manuscript is a resubmission of an earlier submission. The following is a list of the peer review reports and author responses from that submission.

Round 1

Reviewer 1 Report

Comments and Suggestions for Authors

Congratulations, your paper is very interesting and very much relevant for the literature on the topic and for Douro region and Portugal.

Nevertheless, your paper presents improvement opportunities as follows:

1. Introduction – despite you referring missing literature, you need to discuss better your expected contributions compared to those in related (referred) papers (even if indirectly).

2. CO2 & Literature review. You focus a lot on the footprint issues, but you barely do not have any literature on this (only 1 explicit). Furthermore, most of literature used is not recent, not reflecting currents trends in this issue.  

3. You use lots of figures, which is welcomed, but not so much discussion about. Can you please reinforce these analyses (particularly, from Fig.3 to 10)?

4. Add the limitations of your research.

Many thanks

Comments on the Quality of English Language

In general, English is OK, but polishing is required (e.g. The first use of DDR is not explained)

Furthermore, do please pay attention to words in Portuguese as, e.g. Figure 1, or "Custo" (Fig. 11).

Reviewer 2 Report

Comments and Suggestions for Authors

Dear Authors,

The Authors show an exclusive approach to viticulture, which makes it unique in the context of work on the Wine LCA in DDR. With collection of primary data, which is accessible and measurable in the day- to-day life of a vine-growing farm, it is possible to extract economic and environmental results with a high degree of precision and reliability. The detail applied to the retrieval of primary data is crucial if the potential for extracting results is to be effective. From there, it is possible to use universally accessible and open methodologies for calculating economic and environmental impacts, such as AgriBalyse available on OpenLCA, which ensure that the results obtained are reliable and can be compared with each other.

The description of the work is acceptable. Overall impression is that this manuscript can be recommended for publication after MAJOR revision in Sustainability especially considering the scope and topics of this journal. However, I would like to point out to several details:

  1. The data of methods that include in this paper is dependent on the matrix. That effect is very important in the real samples but the authors did not explain the effect of the matrix. Correct and explain this.
  2. It is not clear what novelty in paper worth to publish is? Correct this.
  3. The Figures caption must consist of more details. Correct this.
  4. There is no enough data about the use of PPP (plant protection products), records and pest monitoring. This is VERY IMPORTANT part which includes prevention and/or suppression, monitoring, decision making, non-chemical methods, pesticide selection, reduced pesticides use, anti-resistance strategies and evaluation. Some of this section are mandatory and could find in Field book. Explain this.
  5. Authors give a lot of cost calculations and other calculations but what about the current state of knowledge on models to define and to understand fate and behavior of a pesticide active ingredient and to predict concentration in soil, water and air. Explain and correct this.
  6. English language should be corrected by a professional lector. A proof reading by a native English speaker should be conducted to improve both language and organization quality.

I wish a lot of success to the authors in making this manuscript much better.

With kind regards!

Reviewer 

Comments on the Quality of English Language

English language should be corrected by a professional lector. A proof reading by a native English speaker should be conducted to improve both language and organization quality.

Reviewer 3 Report

Comments and Suggestions for Authors

Based on substantial empirical data and analysis, the author thoroughly explores sustainability issues within grape cultivation. I have no negative opinions regarding this aspect. In terms of writing, I would encourage the author to consider enhancing the coherence of the article, aiding readers in better understanding the intended viewpoints. My specific suggestions are as follows:

Adding a closing paragraph in the 'Introduction' section to briefly outline the structure of the article.

Explicitly stating the novel contributions of this work and highlighting its similarities and differences compared to previously published papers.

Clarifying the academic and practical significance of this study. I recommend including a paragraph in the 'Discussion' section to address its significance.

To swiftly convey the contributions, it would be beneficial to clearly emphasize in the abstract and introduction the primary challenges faced and the original outcomes achieved in overcoming these challenges.

How should future research delve further into this field?

Reviewer 4 Report

Comments and Suggestions for Authors

The organization of this work is very distracting. Authors need to organize paper by using the sequence: introduction, literature review, method, results, discussion and conclusion.

Authors also need to present the research gap. Why this paper is worthwhile.  

Authors also need to explain the figure one by one as well as the main point and implication of the figure instead of presenting the results itself. It should be presented as more convinced manners. Currently, the main point is hard to figure out. 

In general, this paper is very poorly organized.